# Sex differences in health-related quality of life trajectories following myocardial infarction: national longitudinal cohort study

Tatendashe Bernadette Dondo [1,2] Theresa Munyombwe [1,2] Marlous Hall,[1,2] Ben Hurdus,[2,3] Anzhela Soloveva,[1,2] Gerard Oliver,[4] Suleman Aktaa,[1,3] Robert M West [5] Alistair S Hall,[1,3] Chris P Gale[1,2,3]

¹Leeds Institute of Cardiovascular and Metabolic Medicine, University of Leeds, Leeds, UK
²Leeds Institute for Data Analytics, University of Leeds, Leeds, UK
³Department of Cardiology, Leeds General Infirmary, Leeds, West Yorkshire, UK
⁴Patient representative, Lancashire, UK
⁵Leeds Institute of Health Sciences, University of Leeds, Leeds, UK

**Correspondence to**
Dr Theresa Munyombwe;
T.Munyombwe@leeds.ac.uk

## ABSTRACT

**Objectives** To investigate sex-based differences in baseline values and longitudinal trajectories of health-related quality of life (HRQoL) in a large cohort of myocardial infarction (MI) survivors after adjusting for other important factors.

**Design** Longitudinal cohort study.

**Setting** Population-based longitudinal study the Evaluation of the Methods and Management of Acute Coronary Events study linked with national cardiovascular registry. Data were collected from 77 hospitals in England between 1 November 2011 and 24 June 2015.

**Participants** 9551 patients with MI. Patients were eligible for the study if they were ≥18 years of age.

**Primary and secondary outcome measures** HRQoL was measured by EuroQol five-dimension, visual analogue scale (EQ-5D, EQ VAS) survey at baseline, 1, 6 and 12 months after discharge. Multi-level linear and logistic regression models coupled with inverse probability weighted propensity scoring were used to evaluate sex differences in HRQoL following MI.

**Results** Of the 9551 patients with MI and complete data on sex, 25.1% (2,397) were women. At baseline, women reported lower HRQoL (EQ VAS (mean (SD) 59.8 (20.4) vs 64.5 (20.9)) (median (IQR) 60.00 (50.00–75.00) vs 70.00 (50.00–80.00))) (EQ-5D (mean (SD) 0.66 (0.31) vs 0.74 (0.28)) (median (IQR) 0.73 (0.52–0.85) vs 0.81 (0.62–1.00))) and were more likely to report problems in each HRQoL domain compared with men. In the covariate balanced and adjusted multi-level model sex differences in HRQoL persisted during follow-up, with lower EQ VAS and EQ-5D scores in women compared with men (adjusted EQ VAS model sex coefficient: −4.41, 95% CI −5.16 to −3.66 and adjusted EQ-5D model sex coefficient: −0.07, 95% CI −0.08 to −0.06).

**Conclusions** Women have lower HRQoL compared with men at baseline and during 12 months follow-up after MI. Tailored interventions for women following an MI could improve their quality of life.

**Trial registration number** ClinicalTrials.gov (NCT04598048, NCT01808027, NCT01819103)

## INTRODUCTION

Recent decades were characterised by significant decline of mortality in myocardial infarction (MI). Consequently, health-related quality of life (HRQoL) following MI emerged as another important indicator of patient care. HRQoL represents patients' perspective of their health state but also serves as an important clinical risk marker and treatment target given lower HRQoL in MI survivors is independently associated with increased risk of death.[1]

Emerging evidence points to the significant sex-based differences in MI population that may also account for HRQoL differences. The exact explanation for this phenomenon remains uncertain, but distinct clinical presentation and aetiology of MI, higher age and comorbidity burden, less frequent invasive therapeutic approach, higher rehospitalisation rates and long-term mortality had been consistently shown in women compared with men.[2–5] Importantly, these differences in characteristics and treatment strategies may impact not only HRQoL at the time of the acute event but also its trajectories over time. Previous studies that demonstrated lower HRQoL scores in women compared with men were either small[6–10] or focused on a selected subgroup of patients with MI[11 12] thus were

### STRENGTHS AND LIMITATIONS OF THIS STUDY

⇒ Data source is linked nationwide longitudinal health-related quality of life (HRQoL) data, which minimises selection bias and increases generalisability.
⇒ An inverse weighted propensity scoring approach was applied to weight data and balance out systematic differences based on observed covariates to minimise inherent bias.
⇒ Used generic quality of life metric rather than a disease-specific one to measure HRQoL following myocardial infarction.
⇒ Potential selection bias due to loss to follow-up.

unable to adjust for multiple confounding factors or answer the question of independent sex differences in a heterogeneous MI population. Moreover, only a few contemporary studies explored longitudinal HRQoL estimates depending on sex[6 8–10 12 13] thus an appropriate time for the subsequent assessment of HRQoL remains unknown. Knowledge of such sex-based disparities in HRQoL is important as it could highlight the need for strategies to improve the health status of women following MI. Furthermore, understanding the differences in the HRQoL domains may provide an opportunity to identify the components of patient-reported health that need particularly higher attention and clinical counselling. Using a large nationwide longitudinal cohort study of consecutive patients hospitalised with acute coronary syndrome (ACS), we aimed to investigate sex differences in HRQoL in MI survivals, the longitudinal trajectories of HRQoL over a 12-month period, and determine to what extent sex itself might explain the differences in HRQoL when accounting for other important factors.

## METHODS
### Design and setting
Linked data from the Evaluation of the Methods and Management of Acute Coronary Events (EMMACE 3 and 4)[14] and Myocardial Ischaemia National Audit Project (MINAP)[15] were used for the analyses. The EMMACE studies are multicentre nationwide longitudinal cohort studies of patients hospitalised with ACS. Patients were eligible for the study if they were ≥18 years of age. HRQoL data for MI survivors from 77 hospitals in England between 1 November 2011 and 24 June 2015 were collected at hospital admission (baseline), and longitudinally at 1, 6 and 12 months via questionnaires. Patients were consented for data linkage with MINAP to obtain information on the type of MI, baseline co-morbidities and in-hospital treatments. Fifteen participants (0.2%) had missing sex data and were excluded from the study. Of the 9551 participants, 35.7% (3413) completed and returned the questionnaires at all-time points data was collected.

### Assessment of HRQoL
EuroQol five dimension (EQ-5D-3L) questionnaire was used to collect HRQoL data.[16]

The EQ-5D-3L descriptive system and visual analogue scale (EQ VAS) were used in this study because the measures have previously been validated in MI patients and were found to be a valid general HRQoL measurement scale post-MI.[17] Furthermore, this generic measure enables comparison of health problems among patients in different National Quality Registries, to understand the overall severity of problems experienced by patients with different diseases and treatment pathways.[18]

The EQ-5D questionnaire consists of questions covering five health domains, which include mobility, self-care, usual activities, pain/discomfort and anxiety/depression. An EQ-5D single score is derived based on these five dimensions taking into account societal preference weights.[19] The EQ-5D-3L profiles for each patient were combined with health state preference values from the UK general population to give EQ-5D-3L health state index scores ranging from −0.5 to 1, with scores less than 0 indicating states 'worse than death', 0 indicating no quality of life or 'death' and 1 indicating full health and therefore no problems in any domain. The index score has been standardised to the UK population and validity of the questionnaire in MI patients has been determined.[16 19 20] The questionnaire also has a visual analogue scale (EQ VAS) that allows participants to rate their current health state. The EQ VAS score ranges from 0 to 100 with 0 denoting the worst imaginable health state and 100 the best imaginable health state. A difference in the score of 7 for EQ VAS and 0.05 for EQ-5D is regarded as the minimal clinically important difference.[21]

### Statistical analyses
Differences in baseline characteristics for men and women were described using frequencies and proportions for categorical data, means and SD for normally distributed continuous data and medians and interquartile ranges (IQR) for non-normally distributed data. Multilevel linear regression was used to assess sex differences in HRQoL (EQ-5D and EQ VAS scores) in MI survivors. As the HRQoL data consisted of repeated measures nested within individuals and individuals nested within hospitals, the multi-level approach was implemented. Inverse probability weighted propensity scoring was used to weight the data and balance out systematic differences in baseline characteristics between men and women to minimise selection bias (see online supplemental methods section 1, which gives further detail of the methods). The primary outcomes of the study were the EQ-5D and EQ VAS scores—with further subgroup analyses conducted for each of the EQ-5D domains (mobility, self-care, usual activities, pain/discomfort and anxiety/depression) using multilevel logistic regression models. The domains are recorded as three level variables, however, for this study, they were treated as binary variables, 'some problems' and 'extreme problems' levels versus 'no problems'. The 'extreme problem' category of the EQ-5D measure was endorsed by few individuals for some domains (eg, self-care and mobility), therefore, we combined the EQ-5D levels 'some problems' and 'extreme problems'.

To mitigate residual confounding, the multilevel linear and logistic regression models were adjusted for covariates which included aspirin, β-blockers, statins, angiotensin-converting enzyme inhibitor or angiotensin receptor blocker and P2Y$_{12}$ inhibitors prescription at hospital discharge, type of MI, enrolment into cardiac rehabilitation, coronary intervention, body mass index (BMI), previous MI, age, index of multiple deprivation (IMD) score, previous coronary artery bypass graft (CABG) surgery, smoking status, previous percutaneous coronary intervention (PCI), family history of coronary

heart disease, peripheral vascular disease, hypercholesterolemia, previous angina, chronic obstructive pulmonary disease (COPD) or asthma, diabetes mellitus, chronic renal failure, hypertension, cerebrovascular disease and heart failure. Effect sizes were estimated evaluating changes in HRQoL over time, that is, from time of hospitalisation with MI (baseline) to 12 months post hospital discharge. An interaction term of time and sex was added to the models to test if there were significant sex differences in rate of improvement in HRQoL following AMI. The models were fitted on the weighted balanced data.

Multiple imputation by chained equations[22] was used to impute missing data for the following variables: age, IMD score and BMI (see online supplemental table 1), section 2, which gives detail of the imputation strategy used). Based on clinical expert opinion select binary treatment and medical history variables were imputed to 'no' if missing (See online supplemental table 1), section 2, which gives detail of the imputation strategy used).[22] Rubin's rules[23] were used to pool the results' estimates and generate 95% CIs. On non-weighted data predictors of change in HRQoL were explored by sequentially adding covariates (baseline HRQoL patient-reported measures and patient baseline characteristics) to the bivariate multilevel linear regression model with sex only. Covariates which attenuated the sex differences in change in HRQoL observed were considered as predictors. Analyses were performed using Stata MP64 V.14 (StataCorp,www.stata.com) and R V.3.1.2. P-values <0.05 were considered statistically significant.

### Patient and public involvement
The Leeds Teaching Hospitals NHS Trust Cardiovascular Patient and Public Involvement group was involved in the project design. We also worked closely with a patient (GO) outside the group for the interpretation of the research findings, critical review of the manuscript and its dissemination.

## RESULTS
### Study sample
At baseline (admission), a total of 9551 patients with MI and complete data on sex, 25.1% (2397) were women and the average age of the sample was 64.1 years, SD (11.95).

A total of 3413 had complete follow-up data at all-time points, 24.6% (841) were women and 75.36% (2572) were men. Characteristics of patients with missing follow-up data at one or more time points versus those with complete follow-up data at all-time points are presented in online supplemental table 2). Patients with missing follow-up were younger, more likely to live in deprived areas as shown by higher IMD score, more likely to have more risk factors and comorbidities (smoking, diabetes, heart failure, chronical renal failure, COPD, previous AMI).

At baseline, compared with men, women were older (mean age 67.1 (SD 12.0) years vs 63.1 (SD 11.7) years), more likely to have hypertension (51.6 vs 42.7%), COPD/

asthma (16.3 vs 11.7%), and to present with non-ST-elevation myocardial infarction (62.9 vs 57.9%) (table 1). Conversely, men were more frequently smokers (68.9 vs 62.2%), had higher rates of previous MI (17.6 vs 14.3%), previous PCI (10.5 vs 8.1%), or previous CABG surgery (7.9 vs 4.7%) and were more likely to undergo coronary intervention during the hospital stay (48.6 vs 41.6%) compared with women.

### Patterns of HRQoL
At baseline, women had lower HRQoL compared with men: EQ VAS mean (SD) 59.8 (20.4) and versus 64.5 (20.9), EQ VAS median (IQR) 60.00 (50.00–75.00) versus 70.00 (50.00–80.00) and EQ-5D mean (SD) 0.66 (0.31) and versus 0.74 (0.28), EQ-5D median (IQR) 0.73 (0.52–0.85) versus 0.81 (0.62–1.00). The observed difference persisted through all-time points of follow-up (figure 1). Over time HRQoL improved for both men and women following MI (figure 1). Compared with men, women were more likely to report problems in all dimensions of EQ-5D (figure 2). In the first month for both men and women, there was an increase in the proportion of patients reporting problems with usual activities, pain/discomfort and anxiety/depression (figure 2). However, improvements were observed in the following months, with proportions of patients reporting pain/discomfort remaining stagnant (figure 2).

### Adjusted sex differences in HRQoL
The standardised differences showed that the weighting using the propensity scores balanced the systematic differences in baseline characteristics between men and women as the standardised differences were close or equal to zero (see online supplemental table 3, which gives detail of the standardised differences). The minimum propensity score for each level was sufficiently >0 and that the maximum propensity score for each level was sufficiently <1, showing that the overlap assumption was not violated (online supplemental figure 1). Compared with men, women had on average a lower HRQoL (adjusted EQ VAS model sex coefficient: −4.41, 95% CI −5.16 to −3.66 and adjusted EQ-5D model sex coefficient: −0.07, 95% CI −0.08 to −0.06) and higher odds of reporting problems across all individual EQ-5D dimensions (table 2). The interaction term exploring sex-based differences in the rates of HRQoL changes was not significant.

### Factors associated with sex differences in HRQoL
Sex differences were observed in HRQoL in the bivariate model (EQ VAS model sex coefficient: −3.78, 95% CI −4.65 to −2.91 and EQ-5D model sex coefficient: −0.07 to −0.08 to −0.06) (table 3). The sex effect was markedly attenuated after accounting for patients' baseline HRQoL scores (EQ VAS coefficient: −2.56, 95% CI −3.38 to −1.73) (table 3). However, for EQ-5D, baseline scores did not attenuate the sex effect observed.

**Table 1** Patient baseline characteristics, stratified by sex

| Variables | Men n=7154 | Women n=2397 | P value | Missing* |
|---|---|---|---|---|
| NSTEMI, n (%) | 4141 (57.9) | 1507 (62.9) | <0.001 | 0 |
| Age, mean (SD), year | 63.1 (11.7) | 67.1 (12.0) | <0.001 | 19 (0.2) |
| White ethnicity, n (%) | 6027 (96.9) | 2099 (98.5) | <0.001 | 1197 (12.5) |
| IMD, median (IQR) | 17.9 (10.7–31.4) | 20.5 (11.8–33.4) | <0.001 | 5258 (55.1) |
| BMI, mean (SD), kg/ m$^2$ | 28.6 (6.1) | 28.9 (6.0) | 0.151 | 3366 (35.2) |
| Previous angina, n (%) | 1339 (19.7) | 451 (19.9) | 0.826 | 493 (5.2) |
| Diabetes, n (%) | 1256 (18.2) | 457 (19.6) | 0.133 | 329 (3.4) |
| Hypertension, n (%) | 2904 (42.7) | 1170 (51.6) | <0.001 | 487 (5.1) |
| Heart failure, n (%) | 150 (2.2) | 62 (2.7) | 0.151 | 503 (5.3) |
| Peripheral vascular disease, n (%) | 238 (3.6) | 79 (3.6) | 0.992 | 626 (6.6) |
| Cerebrovascular disease, n (%) | 305 (4.5) | 123 (5.4) | 0.065 | 496 (5.2) |
| Chronical renal failure, n (%) | 203 (3.0) | 86 (3.8) | 0.057 | 499 (5.2) |
| COPD/asthma, n (%) | 796 (11.7) | 370 (16.3) | <0.001 | 429 (4.5) |
| Smoker and ex-smoker, n (%) | 4786 (68.9) | 1456 (62.2) | <0.001 | 263 (2.8) |
| CABG surgery, n (%) | 534 (7.9) | 107 (4.7) | <0.001 | 496 (5.2) |
| Previous PCI, n (%) | 713 (10.5) | 184 (8.1) | <0.001 | 510 (5.3) |
| Previous MI, n (%) | 1196 (17.6) | 324 (14.3) | 0.0003 | 486 (5.1) |
| Cardiac rehabilitation† (n=9307), n (%) | 6387 (97.7) | 2110 (97.5) | 0.565 | 607 (6.4) |
| Coronary intervention† (n=8859) (PCI/CABG), n (%) | 2810 (48.6) | 826 (41.6) | <0.001 | 1094 (12.4) |
| Discharge medications† | | | | |
| Beta-blocker (n=8029), n (%) | 5691 (98.4) | 1888 (98.0) | 0.166 | 322 (4.0) |
| ACE or ARB inhibitor (n=8134), n (%) | 5727 (97.8) | 1871 (97.0) | 0.051 | 348 (4.3) |
| Statin (n=8520), n (%) | 6118 (99.1) | 2009 (98.9) | 0.265 | 317 (3.7) |
| Aspirin (n=8499), n (%) | 6107 (99.4) | 2026 (99.0) | 0.048 | 308 (3.6) |
| P2Y$_{12}$ inhibitors (n=5491), n (%) | 3610 (97.6) | 1259 (96.5) | 0.037 | 486 (8.9) |

*15 (0.2%) patients had missing sex data.
†Only patients eligible to receive treatments were included in the denominator of the complete cases.
ACEi, angiotensin-converting enzyme inhibitor; ACS, acute coronary syndrome; ARB, angiotensin receptor blocker; BMI, body mass index; CABG, coronary artery bypass graft; COPD, chronic obstructive pulmonary disease; IMD, index of multiple deprivation; MI, myocardial infarction; NSTEMI, non-ST-elevation myocardial infarction; PCI, percutaneous coronary intervention.

## DISCUSSION

In this national longitudinal cohort study of 9551 consecutive patients hospitalised with MI, we demonstrated that (1) women had lower HRQoL compared with men at baseline and throughout the following 12 months; (2) trajectories in HRQoL scores and all EQ-5D-3L domains (mobility, personal care, activities of daily living, pain/discomfort and anxiety/depression) assessed at four time points were similar between groups; and (3) adjustment for other variables, including age, risk factors, comorbidity, treatment, final diagnosis and baseline HRQoL decreased but did not eliminate the differences observed in HRQoL in women and men following MI.

To our knowledge, we present the largest longitudinal study to assess sex differences in MI survivors. Prior studies have addressed this question, however, they have been limited to small sample sizes,[6–10] subselecting only patients with MI receiving certain interventions,[11] and

short follow-up.[11 24] Data on 12 months HRQoL trajectories from contemporary real-world patient populations are limited.

Similar to our findings, previous research has shown that women report lower HRQoL at time of their presentation with MI,[13 25] but the sex differences in baseline health status prior to MI have been attributed to the fact that women usually report more mental health disorders such as depression, fatigue or anxiety compared with men.[26–28] Our study, similarly to a recent large study of contemporary ACS patients treated with PCI, found that female sex was independently associated with significant impairment in all EQ-5D-3L domains (mobility, personal care, activities of daily living, pain/discomfort and anxiety/depression).[11] Moreover, during longitudinal 12-month assessment, women consistently reported lower HRQoL as measured by overall EQ-5D score, EQ VAS and problems at each of EQ-5D domains.

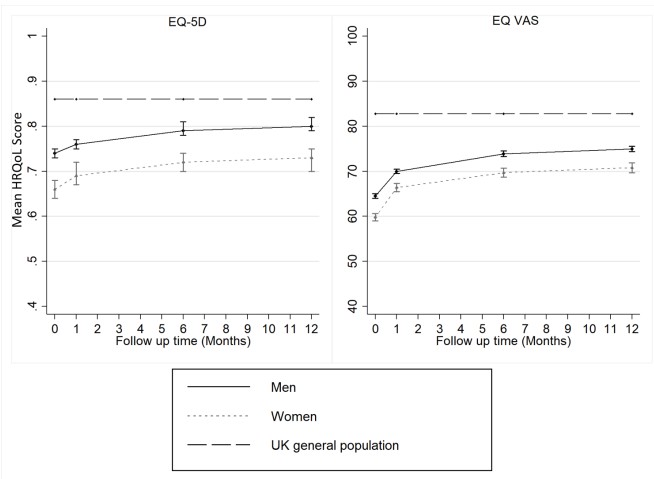

**Figure 1** HRQoL trajectories following myocardial infarction by sex and UK general population. EQ-5D, EuroQol five dimension; EQ VAS, EuroQol visual analogue scale; HRQoL, health-related quality of life.

Between sex differences in epidemiology, pathophysiology, risk factors, clinical presentation and treatment strategies that have been demonstrated for MI patients are likely contributing to the observed differences in HRQoL following MI. Coronary revascularisation after MI was associated with improvements in HRQoL for both men and women, yet similarly to prior findings in our study women less frequently underwent coronary intervention. In recent years, an increasing emphasis has been placed on the association between multimorbidity level and negative outcomes in MI survivors. Indeed, the changes in HRQoL in men have been found to be associated with presenting characteristics of MI and complications of treatment while those of women were linked with their demographic characteristics and comorbidities.[13 29] In our study, women were older and had higher premorbid conditions at baseline, the presence of which has been associated with worst HRQoL following MI. Importantly, however, after adjustment for

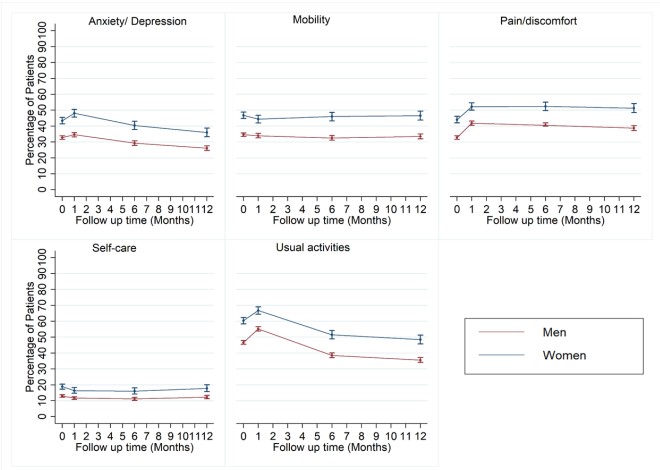

**Figure 2** Health-related quality of life domains trajectories following myocardial infarction by sex.

**Table 2** Propensity score analysis to show health-related quality of life differences between women versus men

| Health-related quality of life | Coefficient (95% CI) | P value |
| --- | --- | --- |
| EQ VAS model | | |
| Sex (women vs men) | −4.41 (−5.16 to −3.66) | <0.001 |
| EQ-5D model | | |
| Sex (women vs men) | −0.07 (−0.08 to −0.06) | <0.001 |
| **EQ-5D dimensions** | **OR (95% CI)** | |
| Mobility problems model | | |
| Sex (women vs men) | 1.82 (1.58 to 2.09) | <0.001 |
| Activities of daily living problems model | | |
| Sex (women vs men) | 1.70 (1.52 to 1.89) | <0.001 |
| Self-care problems model | | |
| Sex (women vs men) | 1.75 (1.47 to 2.08) | <0.001 |
| Pain/discomfort model | | |
| Sex (women vs men) | 1.59 (1.45 to 1.75) | <0.001 |
| Anxiety/depression model | | |
| Sex (women vs men) | 2.03 (1.80 to 2.29) | <0.001 |

EQ-5D, EuroQol five dimension; EQ VAS, EuroQol visual analogue scale.

multiple confounders, including comorbidities and treatment strategies such as medication, revascularisation and cardiac rehabilitation, between sex differences in HRQoL remained significant for our study. Another study though suggested an impact of sex on physical functioning only, while gender-related factors such as femininity score, social support, and housework responsibility are independent predictors of long-term HRQoL.[30] Recognition of the associations of sex and gender itself with a diverse spectrum of factors related to cardiovascular and general health has led to a recently proposed concept of sex-and gender-sensitive medicine. Still translation of this concept into routine clinical care is far from desired. Our study further magnifies this by demonstrating the lack of trend towards closing the between-sex gap during 12 months follow-up after MI in the large clinical practice-based patient population data from the EMMACE studies. From clinical medicine and physicians' perspective, in order to reduce the existing disparities, attempts should be sought to improve women's health status by identification of as many potential reasons as possible, addressing the modifiable risk factors and engaging more women into the recommended multidisciplinary post-MI management programmes. Lack of systematic assessment of gender-specific factors in many studies, including our study, highlight a need for large-scale strategic initiatives from public health and social care to better understand and support many intertwined factors affecting women health.

In our study, the meaningful improvement in HRQoL has been demonstrated equally in both men and women. Although our study is in line with a previous analysis

**Table 3** Factors explaining sex differences observed in HRQoL following MI

| Parameter | EQ-5D model, coefficient (95% CI) | EQ VAS model, coefficient (95% CI) |
|---|---|---|
| Sex effect | −0.07 (−0.08 to −0.06) | −3.78 (−4.65 to −2.91) |
| Adding age, BMI, IMD | −0.06 (−0.07 to −0.05) | −3.33 (−4.20 to −2.40) |
| Adding pharmacotherapy and coronary intervention | −0.06 (−0.08 to −0.05) | −3.27 (−4.16 to −2.27) |
| Adding final diagnosis | −0.06 (−0.08 to −0.05) | −3.17 (−4.06 to −2.27) |
| Adding comorbidities and risk factors* | −0.07 (−0.08 to −0.06) | −3.77 (−4.64 to −2.90) |
| Adding baseline value of the HRQoL metric† | −0.06 (−0.07 to −0.05) | −2.56 (−3.38 to −1.73) |

*Baseline EQ-5D for EQ-5D model and baseline EQ VAS for EQ VAS model.
†Previous MI, age, previous coronary artery bypass graft surgery, smoking status, previous percutaneous coronary intervention, family history of coronary heart disease, peripheral vascular disease, hypercholesterolemia, previous angina, chronic obstructive pulmonary disease or asthma, diabetes mellitus, chronic renal failure, hypertension, cerebrovascular disease and heart failure.
BMI, body mass index; EQ-5D, EuroQol five dimension; EQ VAS, EuroQol visual analogue scale; HRQoL, health-related quality of life; IMD, index of multiple deprivation; MI, myocardial infarction.

of young MI populations in terms of general positive HRQoL trend after MI independently of sex,[31] in other studies, different patterns have been found for women mainly reporting an improvement in mental functioning while men tend to report improvement in the physical health status.[8 32] Analysis of magnitude of change in health status showed the highest increment improvements of EQ-5D and EQ VAS score at 1 month after discharge and it is plateauing after 6 months. Counterintuitively, however, at 1 month time point, the highest proportions of patients reported some problems in anxiety/depression, pain/discomfort and usual activities. This highlights that measuring not only health status via serial general HRQoL assessments but also its domains are needed. Moreover, considering independent relationship between EQ-5D and its domains in MI survivors and mortality,[1] this strategy might provide advantages in identification of patients at the highest risk of negative outcomes. Indeed, analysis of 26 641 patients with first MI from SWEDEHEART registry have shown that anxiety/depression assessed 6–10 weeks after MI is associated with 29% higher risk of cardiovascular mortality and 34% higher risk of non-cardiovascular mortality independently of traditional risk factors. Though these associations remained significant only if the mental problems persisted after 12 months.[33] Another multinational study of HRQoL as assessed by EQ-5D in 8978 post-MI patients showed that the presence of problems on 'self-care' and 'mobility' were most powerful predictors of all-cause mortality, whereas problems with pain/discomfort and usual activities were most strongly associated with cardiovascular events.[1]

Our findings can be interpreted using the Wilson Clearly HRQoL[34] conceptual model, which causally links five health concepts, the biological and physiological factors, symptoms, functional health, general health perceptions and HRQoL. Symptoms mediate between physiological factors and functional status; functional status mediates between symptoms and general health perceptions, and general health perceptions mediates between functional status and overall HRQL.[35] A systematic review found that more symptoms implied impaired functioning, which may lead to worse general health perception and consequently lower HRQoL.[35] Compared with men, women reported more symptoms and problems with physiological factors and functional status and these potentially mediate between functional statuses, which can have a direct effect on overall HRQoL.

Future large studies of sex and gender differences and effects of targeted interventions after MI might help to further personalise management strategies and as a result improve HRQoL outcomes in women.

## Implications of the study
Our findings build on other previous studies suggesting lower HRQoL in women compared with men but strengthen them by reporting absence of significant difference in patterns of changes throughout 12 months follow-up in a broad real-world MI population. Higher adoption of serial assessment of patient-reported outcomes such as HRQoL is needed to tailor treatment interventions. The quantification of HRQoL at time of MI and identification of predictors of recovery may be important for designing targeted interventions tailored to meet the needs of patients and improve their physical and mental well-being.

## Strengths and limitations
Our study has strengths in that it evaluates changes in HRQoL using nationwide longitudinal data, which minimises selection bias and increases generalisability. There are no other databases of comparable size, coverage and quality. An inverse-weighted propensity scoring approach was applied to weight data and balance out systematic differences based on observed covariates to minimise inherent bias. However, our study has limitations. (1) We used a generic quality of life metric rather than a disease-specific one to measure HRQoL. Nonetheless, the EQ-5D does capture dimensions of the quality of life that are relevant to, and are impacted

by, MI such as mobility, depression/anxiety and pain/discomfort. In addition, EQ-5D has been validated in patients with MI, and using a generic metric allows the comparison for the magnitude of HRQoL impairment between MI and other diseases. More work is required using causal HRQoL conceptual frameworks such as the Wilson Clearly causal framework[34] to gain a deeper understanding of the nature of HRQoL and factors contributing to it. (2) The generalisability of the study's findings may be limited by a selection bias inherent as a result of loss to follow-up data. However, sensitivity analyses comparing those lost follow-up with those who were not shown minimal systematic differences (see online supplemental table 2, which compares baseline characteristics of patients with complete follow-up with those missing one or more follow-up data points). (3) Similar to many large cardiovascular registries, our study did not collect sex-specific characteristics, particularly social status and mental disorders prior to MI, therefore, there could be residual confounding. However, we adjusted for an extensive range of patient-level factors that are usually included in similar sex-based research including IMD, which is a measure of relative deprivation derived from combining seven domains: income, employment, education, skills and training, health and disability, crime to housing services and living environment.

## CONCLUSION

In this national longitudinal study, women have lower HRQoL by index EQ-5D score at baseline and during 12-month follow-up after MI and persistently reported higher levels of impairment in their mobility, self-care, usual activities, pain/discomfort and anxiety/ depression at each time point assessment. The magnitude of HRQoL improvement was similar between groups. Sex differences were attenuated by baseline HRQoL scores. Targeted interventions to address the reasons behind poor baseline health status, particularly gender-specific factors and multiple domains of HRQoL, could improve health outcomes in women after MI. During cardiac rehabilitation following MI, EQ-5D can be used as a tool to identify women at risk of poor HRQoL and dimensions mostly affected allowing targeted intervention.

**Acknowledgements** We gratefully acknowledge the contributions from all hospitals, healthcare professionals and patients who participated in the EMMACE study. We would also like to thank Richard Gillott (IT support), Claire Forrest (Cardiology Research Coordinator) and Vera Hall (Finance support).

**Contributors** TBD analysed the data and drafted the manuscript. CG and ASH contributed to the conception of the research, funding acquisition, project administration, supervision, study design and data collection. CG, ASH, SA, AS and BH provided expert clinical opinion and interpretation of the data. RMW, MH, TM and TBD provided statistical expert advice and interpretation of the data. GO was involved as a patient advisor in the interpretation of the research and the writing of the manuscript. All authors made critical revisions and provided intellectual content to the manuscript, approved the final version to be published and agreed to be accountable for all aspects of the work.TBD acts as guarantor of this manuscript.

**Funding** This work was supported by the National Institute for Health Research (NIHR/CS/009/004) and British Heart Foundation (PG/19/54/34511). MH was funded by the Wellcome Trust (grant reference: reference: 206470/Z/17/Z).

**Competing interests** CG reports personal fees from AstraZeneca, personal fees from Bayer, personal fees from Boehringer Inglehiem, personal fees from Amgen, personal fees from Daiichi Sankyo, personal fees from Vifor Pharma, grants from Abbott, grants from BMS, outside the submitted work. BH reports grants from National Institute for Health Research (NIHR/CS/009/004) and British Heart foundation (PG/19/54/34511), during the conduct of the study. AH reports personal fees (speaker honorarium) from NOVARTIS & SERVIER. AS acknowledges funding received from the European Society of Cardiology in form of an ESC Research Grant.

**Patient and public involvement** Patients and/or the public were involved in the design, or conduct, or reporting, or dissemination plans of this research. Refer to the Methods section for further details.

**Patient consent for publication** Not applicable.

**Ethics approval** EMMACE-3 has been given a favourable ethical opinion by the Leeds (Central) Research Ethics committee (REC reference: 10/H1313/74), is registered on ClinicalTrials.gov (NCT01808027) and has been adopted by the National Institute for Health Research Comprehensive Research Network portfolio (9102). EMMACE-4 has been given favourable ethical opinion by the West Midlands—Black Country Research Ethics Committee (REC reference: 12/WM/0431), is registered on ClinicalTrials.gov (NCT01819103) and has been adopted by the National Institute for Health Research Comprehensive Research Network portfolio (9102). Participants gave informed consent to participate in the study before taking part.

**Provenance and peer review** Not commissioned; externally peer reviewed.

**Data availability statement** Data are available upon reasonable request.

**ORCID iDs**
Tatendashe Bernadette Dondo http://orcid.org/0000-0002-8337-8425
Theresa Munyombwe http://orcid.org/0000-0002-1307-6691
Robert M West http://orcid.org/0000-0001-7305-3654

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
