## [Reviewer comments · BMJ Open]

ARTICLE DETAILS

TITLE (PROVISIONAL)	Sex differences in health-related quality of life trajectories following myocardial infarction: National longitudinal cohort study.
AUTHORS	Dondo, Tatendashe; Munyombwe, T; Hall, Marlous; Hurdus, B; Soloveva, A.; Oliver, Gerard; Aktaa, Suleman; West, Robert; Hall, Alistair; Gale, Chris

VERSION 1 – REVIEW

REVIEWER	Kang, Kyoungrim Pusan National University, College of Nursing
REVIEW RETURNED	24-Mar-2022

GENERAL COMMENTS	Thank you for the opportunity to review this well-written manuscript. I only have a few minor suggestions. - As HRQOL was the major concept of this manuscript, the reason that the authors selected the generic tool for assessment should be addressed in the main text, either in methods or in discussion, apart from the limitations.- Compared to adequate results and discussion, conclusions of this study seemed weak. Please provide more directions for the future study based on the characteristics of this target population.- Some references look outdated. To reflect more of the current circumstance, please provide more recent references.
--

REVIEWER	Wlodarczyk, Dorota Medical University of Warsaw
REVIEW RETURNED	23-Apr-2022

GENERAL COMMENTS	The aim of the study was to investigate sex-based differences in health-related quality of life (HRQoL) during 12 months follow up after MI in a large cohort and after adjusting for other important factors. The study seems very promising but there are some aspects demanding further explanation. Additionally, in the light of existing data on sex- and gender- differences in HRQoL after MI one may expect that using longitudinal cohort study would bring deeper insight in the problem presented in the study. I would consider the following issues: Introduction - throughout the whole manuscript it is not clear for me how the Authors refer to the issues of sex-based and gender-based HRQoL differences in MI survivals (although the title indicates clearly focus on sex-based differences, in the introduction and discussion sections also gender issue is introduced).
--

	 - it would be recommended to refer to the theoretical models of HRQoL which give deeper insight into the nature of HRQoL and factors contributing to it. - the novelty or added value of the study should be stated more clearly (it seems that in the light of existing data the size of the sample and the number of confounders are not enough). Methods  - As for data collection: when exactly was the baseline measurement conducted (this would be important when interpreting the attenuation of sex effect on HRQoL after controlling for it) and how data on consecutive timepoints were collected? (the way of filling out the questionnaires); what types of sexes were included in the survey (was there any risk to collect false data due to lack of the proper response category for non-binary people?); what is IMD (index of multiple deprivation)? - there was a noticeable drop out in the study (64.3%) – the differences between the baseline group and those participating in all measurements should be described in the main text and discussed in limitations; - as for covariates: I could not find data on the length of hospital stay, education level, marital status and professional status and its change during one year after MI (which are the proved determinants of HRQoL after MI) and mental disorders prior to MI - as described enrolment into cardiac rehab was controlled for but what kind (in- or outpatient) and what about its continuation (in any form) during this year? - as for assessment of HRQoL: why the EQ-5D-3L instead of EQ-5D-5L was chosen? (the latter gives more insightful categorization of health status and its domains); how was an EQ-5D score calculated?; why responses were reduced from 3 levels to two categories? - what was patients' contribution to the interpretation of the research findings, critical review of the manuscript and its dissemination – the short comment would be valuable Results  - the sample description including drop-out and gender differences in this respect should be added - in table 1 – some categorization of factors would enable its reading e.g. clinical presentation and treatment of MI, comorbidity etc. - Figure 1 – data on UK general population should be stratified by sex/gender as same as EQ-5D Discussion The results should be discussed in the context of the broader theoretical model of HRQoL. Otherwise, the value of research is limited to describing reality without contributing to the advancement of knowledge about HRQoL. Discussion should be extended by inclusion the role of the above mentioned factors (education, professional status ect.). More emphasis should be put on sex and gender associations in the context of HRQoL after MI. Also section regarding limitation should be more critical, including relatively short time of observation and noticeable drop out.
--	---

VERSION 1 – AUTHOR RESPONSE

Reviewer: 1

Dr. Kyoungnim Kang, Pusan National University Comments to the Author:

Thank you for the opportunity to review this well-written manuscript. I only have a few minor suggestions.

- As HRQOL was the major concept of this manuscript, the reason that the authors selected the generic tool for assessment should be addressed in the main text, either in methods or in discussion, apart from the limitations.

We have added the following text to the manuscript methods section, page 6-7, lines 144-150. "The EQ-5D-3L descriptive system and EQ VAS were used in this study because the measures have previously been validated in MI patients and were found to be a valid general HRQoL measurement scale post MI¹⁷. Furthermore, this generic measure enables comparison of health problems among patients in different National Quality Registries, to understand the overall severity of problems experienced by patients with different diseases and treatment pathways¹⁸."

- Compared to adequate results and discussion, conclusions of this study seemed weak. Please provide more directions for the future study based on the characteristics of this target population.

We thank the reviewer for the suggestion. We have added to the conclusion of the manuscript the following text to the manuscript "During cardiac rehabilitation following MI, EQ-5D can be used as a tool to identify women at risk of poor HRQoL and dimensions mostly affected allowing targeted intervention." Page 19, lines 422-424.

- Some references look outdated. To reflect more of the current circumstance, please provide more recent references.

Additional contemporary references have now been included, please see references 18, 35.

Reviewer: 2

Dr. Dorota Włodarczyk, Medical University of Warsaw Comments to the Author:

The aim of the study was to investigate sex-based differences in health-related quality of life (HRQoL) during 12 months follow up after MI in a large cohort and after adjusting for other important factors. The study seems very promising but there are some aspects demanding further explanation. Additionally, in the light of existing data on sex- and gender- differences in HRQoL after MI one may expect that using longitudinal cohort study would bring deeper insight in the problem presented in the study. I would consider the following issues:

Introduction

- throughout the whole manuscript it is not clear for me how the Authors refer to the issues of sex-based and gender-based HRQoL differences in MI survivals (although the title indicates clearly focus on sex-based differences, in the introduction and discussion sections also gender issue is introduced).

We thank the reviewer for this important point. We understand that these two are different concepts but cannot be considered separately. The focus of this study is on sex differences as particular gender-specific factors were not collected. We have corrected the manuscript and hope it has improved as a result.

- it would be recommended to refer to the theoretical models of HRQoL which give deeper insight into the nature of HRQoL and factors contributing to it. An understanding of the relationship between HRQOL and factors guided by a theoretical framework is important.

We agree with the reviewer's suggestion of referring to a theoretical model of HRQoL to give deeper insight into the nature of HRQoL and factors contributing to it. However, the current study is restricted by data availability to focus on a deeper insight into the nature of HRQoL and factors contributing to it. Data available on HRQoL is that collected using EQ-5D-3L, a generic tool. This has been highlighted as a limitation of the study in the manuscript as follows, See discussion section, page 18, lines 395-398.

“We used a generic quality of life metric rather than a disease-specific one to measure HRQoL. Nonetheless, the EQ-5D does capture dimensions of the quality of life that are relevant to, and are impacted by, MI such as mobility, depression/anxiety and pain/discomfort. In addition, EQ-5D has been validated in MI patients, and using a generic metric allows the comparison for the magnitude of HRQoL impairment between MI and other diseases. More work is required using causal HRQoL conceptual frameworks such as the Wilson Clearly causal framework to gain a deeper understanding of the nature of HRQoL and factors contributing to it”

- the novelty or an added value of the study should be stated more clearly (it seems that in the light of existing data the size of the sample and the number of confounders are not enough).

We believe that our study is large enough to estimate over 300 parameters with 100% power as detailed by the power calculations below: For example, a sample size of 9551 allows estimation of 300 parameters using regression analysis models.

Using a sample size of 9551, assuming a significance level of 0.05, adjusted R squared of 0.7, estimating 300 parameters, has a power of 100%. Lowering adjusted R squared to 0.2 will still allow us to estimate 300 parameters with 100% power .

The number of parameters estimated in this study are less than 300 using a sample size of 9511.

Methods

- As for data collection: when exactly was the baseline measurement conducted (this would be important when interpreting the attenuation of sex effect on HRQoL after controlling for it) and how data on consecutive time points were collected? (the way of filling out the questionnaires); what types of sexes were included in the survey (was there any risk to collect false data due to lack of the proper response category for non-binary people?); what is IMD (index of multiple deprivation)?

Baseline measurements were taken during admission to hospital. The information used to define sex of the patients used for this study is that which was collected at baseline. The variable was coded with 3 categories, male, female and unknown in the dataset. For the purpose of this study we restricted the analysis to patients who declared either being male or female at admission. Unknown would have defined patients who did not complete or had missing data on sex, so the data entry clerk would enter unknown. We agree with the reviewer that there could be a restriction due to lack of other response categories besides male or female. However, we would not consider the data collected false data.

IMD is index of multiple deprivation, is a measure of relative deprivation derived from combining 7 domains: income, employment, education, skills and training, health and disability, crime to housing services, living environment. The higher the score the more deprived the area the individual lives in.

- there was a noticeable drop out in the study (64.3%) – the differences between the baseline group and those participating in all measurements should be described in the main text and discussed in limitations;

We have added this information to the “Strengths and Limitations” section, lines 402-406, which reads “The generalisability of the study’s findings may be limited by a selection bias inherent as a result of loss to follow-up data. However, sensitivity analyses comparing those lost follow-up with those who were not showed minimal systematic differences (See Supplementary Table 2, which compares baseline characteristics of patients with complete follow-up with those missing one or more follow-up data points).”

We now further describe in text the characteristics of patients with missing follow-up data at one or more time points versus those with complete follow up data at all-time points presented in See Supplementary Table 2. The following text was added in the results section in the main text

‘Characteristics of patients with missing follow-up data at one or more time points versus those with complete follow up data at all-time points are presented in Supplementary Table 2. Patients with missing follow up data were younger, more likely to live in deprived areas as shown by higher IMD score, more likely to have more risk factors and comorbidities (smoking, diabetes, heart failure, chronic renal failure, COPD, Previous AMI).’ See page 9, lines 222-227.

- as for covariates: I could not find data on the length of hospital stay, education level, marital status and professional status and its change during one year after MI (which are the proved determinants of HRQoL after MI) and mental disorders prior to MI

We have highlighted in the discussion pages 18, lines 406-412 that: "Our study used observational data and there was no data on some other confounders, particularly social status and mental disorders prior to MI, therefore there could be residual confounding. However, we adjusted for an extensive range of patient level factors that are usually included in similar sex based research including IMD which is a measure of relative deprivation derived from combining 7 domains: income, employment, education, skills and training, health and disability, crime to housing services, living environment".

- as described enrolment into cardiac rehab was controlled for but what kind (in- or outpatient) and what about its continuation (in any form) during this year?

Enrolment into cardiac rehab controlled for in the study is information recorded at patient discharge on referral for cardiac rehab.

- as for assessment of HRQoL: why the EQ-5D-3L instead of EQ-5D-5L was chosen? (the letter gives more insightful categorization of health status and its domains

We agree with the reviewer that EQ-5D-5L gives more insight compared to the 3L. However, licencing obtained for EQ-5D when the study was set up in 2011 was for EQ-5D-3L, EQ-5D-5L had been launched in 2009.

-how was an EQ-5D score calculated?

We described in the manuscript that 'The EQ-5D-3L profiles for each patient were combined with health state preference values from the UK general population, to give EQ-5D-3L health state index scores ranging from -0.5 to 1.' Page 7, lines 155-157.

-why responses were reduced from 3 levels to two categories?

3 levels were reduced to two because the "extreme problem" category of the EQ5D measure was endorsed by few individuals for some domains (e.g self-care and mobility) therefore we combined the EQ-5D levels 'some problems' and 'extreme problems'. This explanation has been added to the methods section of the main text, it reads "The domains are recorded as three level variables, however, for this study they were treated as binary variables, 'some problems' and 'extreme problems' levels vs. 'no problems'. The "extreme problem" category of the EQ-5D measure was endorsed by few individuals for some domains (e.g self-care and mobility) therefore we combined the EQ-5D levels 'some problems' and 'extreme problems'." See page 8, lines 180-182.

- what was patients' contribution to the interpretation of the research findings, critical review of the manuscript and its dissemination – the short comment would be valuable

The study findings were disseminated at a Leeds Teaching Hospital Trust (LTHT) PPI meeting, where patients agreed that the problems shown by the EQ5D measure mattered to them and were prevalent in MI survivors and study findings were clinically meaningful.

Results

- the sample description including drop-out and gender differences in this respect should be added

We thank the reviewer for this suggestion. We have added the proportions of women vs. men that had information collected and complete at all-time points to the results section. It reads “A total of 3,413 had complete follow up data at all-time points, 24.6% (841) were females and 75.36% (2,572) men. ”. We feel the information summarised in supplementary Table 2, supplementary file comparing patients who had complete follow-up data vs. those who had missing data on one or more time points gives a reflection on information on patients who dropped out of the study despite stratification by sex. See pages 9-10, lines 219-222.

- in table 1 – some categorization of factors would enable its reading e.g. clinical presentation and treatment of MI, comorbidity etc.

Variables presented in table 1 were collected as categorized as binary (yes/no) which we feel would suffice to be understandable by readers. We could not further categorise to any other definition. Age, BMI and IMD were presented using summary statistics that give a reflection of the distribution of the data. Categorising the variables would lead to loss of information, we feel using the continuous version and summarising using the appropriate summary statistic is the accurate way of presenting the data.

- Figure 1 – data on UK general population should be stratified by sex/gender as same as EQ-5D

In figure 1 we aimed to compare HRQoL of men or women compared to the UK general population. The mean for the UK general population for the purpose of this study was considered the ideal/expected HRQoL to be attained following recovery following MI.

Discussion

The results should be discussed in the context of the broader theoretical model of HRQoL. Otherwise, the value of research is limited to describing reality without contributing to the advancement of knowledge about HRQoL.

In the discussion section, we have discussed our study findings in the context of the Wilson Clearly HRQoL conceptual model as follows:

“Our findings can be interpreted using the Wilson Clearly HRQoL ³⁴ conceptual model, which causally links five health concepts, the biological and physiological factors, symptoms, functional health, general health perceptions and HRQoL. Symptoms mediate between physiological factors and functional status; functional status mediates between symptoms and general health perceptions, and general health perceptions mediates between functional status and overall HRQL ³⁵. A systematic review found that more symptoms implied impaired functioning which may lead to worse general health perception and consequently lower HRQoL³⁵. Compared to men, women reported more symptoms and problems with physiological factors and functional status and these potentially mediate between functional statuses which can have a direct effect on overall HRQoL. See page17, 364-376

Discussion should be extended by inclusion the role of the above mentioned factors (education, professional status ect.).

We have made some changes in the discussion and limitation sections, adding that “similar to many large cardiovascular registries, our study did not collect particular sex-specific characteristics, particularly social status and mental disorders prior to MI, therefore there could be residual confounding. However, we adjusted for an extensive range of patient level factors that are usually included in similar sex based research including IMD which is a measure of relative deprivation derived from combining 7 domains: income, employment, education, skills and training, health and disability, crime to housing services, living environment”. See page 18, lines 406-412.

We agree with the reviewer, that this information is important and should be of priority of future large scale initiatives exploring sex- and gender-differences in cardiovascular medicine. We hope, that the complex measure of relative deprivation derived from combining 7 domains: income, employment, education, skills and training, health and disability, crime to housing services, living environment might partly reduce this limitation”.

More emphasis should be put on sex and gender associations in the context of HRQoL after MI. Also section regarding limitation should be more critical , including relatively short time of observation and noticeable drop out.

We thank the review for the critical appraisal of our work and this very important point. We have added more information on sex and gender associations in the context of HRQoL after MI in the discussion and limitation section, see pages 15-16 lines 327-342.

VERSION 2 – REVIEW

REVIEWER	Kang, Kyoungnim Pusan National University, College of Nursing
REVIEW RETURNED	03-Aug-
GENERAL COMMENTS	All my comments have been adequately addressed.